# Anticancer and Targeting Activity of Phytopharmaceutical Structural Analogs of a Natural Peptide from *Trichoderma longibrachiatum* and Related Peptide-Decorated Gold Nanoparticles

**DOI:** 10.3390/ijms24065537

**Published:** 2023-03-14

**Authors:** Francesca Moret, Luca Menilli, Celeste Milani, Giorgia Di Cintio, Chiara Dalla Torre, Vincenzo Amendola, Marta De Zotti

**Affiliations:** 1Department of Biology, University of Padova, 35131 Padova, Italy; 2Department of Chemical Sciences, University of Padova, 35131 Padova, Italy

**Keywords:** phytochemical, peptaibol, antitumor peptide, breast cancer, peptide-decorated gold nanoparticles

## Abstract

In the large field of bioactive peptides, peptaibols represent a unique class of compounds. They are membrane-active peptides, produced by fungi of the genus *Trichoderma* and known to elicit plant defenses. Among the short-length peptaibols, trichogin GA IV is nonhemolytic, proteolysis-resistant, antibacterial, and cytotoxic. Several trichogin analogs are endowed with potent activity against phytopathogens, thus representing a sustainable alternative to copper for plant protection. In this work, we tested the activity of trichogin analogs against a breast cancer cell line and a normal cell line of the same derivation. Lys-containing trichogins showed an IC_50_ below 12 µM, a peptide concentration not significantly affecting the viability of normal cells. Two analogs were found to be membrane-active but noncytotoxic. They were anchored to gold nanoparticles (GNPs) and further investigated for their ability to act as targeting agents. GNP uptake by cancer cells increased with peptide decoration, while it decreased in the corresponding normal epithelial cells. This work highlights the promising biological properties of peptaibol analogs in the field of cancer therapy either as cytotoxic molecules or as active targeting agents in drug delivery.

## 1. Introduction

Peptide-based therapy is considered a promising alternative to small-molecule and monoclonal antibody therapies, being characterized by fewer off-target effects than the former and economically advantageous over the latter [1,2,3]. Several peptide-based drugs currently on the market are used for the treatment of metabolic diseases [4,5], but they are actively studied also for their anticancer and antimicrobial potential [6,7]. Unfortunately, most of the therapeutic peptides cannot be administered orally since they are easily degraded by proteolytic enzymes [8]. Among natural, bioactive peptides there are peptaibols [9,10], an important class of antimicrobial peptides that show resistance to enzymatic degradation, the ability to interact with phospholipid membranes and a stable helical structure [11,12,13]. Lipopeptaibols are a family of short-length peptaibols featuring an N-terminal long-chain acyl group [14,15]. The progenitor of lipopeptaibols is trichogin GA IV, a weakly amphiphilic peptide of ten amino acid residues, featuring a 1,2-aminoalcohol at its C-terminus and a 1-octanoyl group at its N-terminus [16,17]. Trichogin GA IV is produced by the fungus *Tricoderma longibrachiatum* [18] as a defensive weapon against other microorganisms [19,20]. Over the years, several analogs of trichogin GA IV have been produced and evaluated with the aim of applying them as plant-protection products against fungal phytopathogens [21,22,23,24,25] (Appendix A) and also as antimicrobial agents against Gram+ and Gram− bacteria (*S. aureus*, *S. epidermis*, *P. aeruginosa*, *E. coli*) [26,27]. Recently, specific amino acid substitutions on the peptide backbone allowed us to identify trichogin analogs (e.g., K6-Lol and K6-NH2, Table 1) endowed with potent activity against ovarian cancer and Hodgkin lymphoma [28]. Indeed, it was demonstrated in vitro that both peptides exerted cytotoxicity due to membrane permeabilization in different cancer cell lines, including some tumor cells resistant to doxorubicin or cisplatin treatments. Importantly, notwithstanding a quite comparable cytotoxic effect, K6-Lol and K6-NH2 demonstrated higher and more rapid uptake in cancer cells with respect to normal ones, thus suggesting a potential to selectively interact with malignant tissues in vivo.

In the frame of the same systematic substitution of trichogin GA IV amino acids, in the present work, a dozen water-soluble analogs (Table 1), either known [21,29,30] or newly designed, were efficiently produced by ecofriendly solid-phase peptide synthesis and tested in vitro for anticancer activity and selective interaction against breast cancer cells (MDA-MB-231) with respect to normal cells of the same derivation (MCF-10A). We selected a triple negative breast cancer (TNBC) as in vitro model over other BC subtypes since the frequency of its diagnosis has dramatically increased over the recent years, especially among young women [31,32]. Indeed, the TNBC survival rate remains the lowest with respect to those of other BC subtypes: targeted hormonal therapies are almost ineffective, since TNBC cells lack estrogen receptors (ER), progesterone receptors (PR), and human epidermal growth factor receptor 2 (HER2) [33,34].

While the majority of trichogin GA IV analogs herein described displayed potent anticancer activity in the low micromolar range, four of the screened peptides were able to interact with cancer cells very efficiently but without exerting any cytotoxicity. Those peptide sequences derive from trichogin analogs that have already showed potent membrane interaction and poor cytotoxicity [29]. Thus, in the present study we also report on the possibility to exploit them as targeting agent for the decoration of gold nanoparticles (GNPs) to be used for drug delivery or diagnostic purposes in the TNBC therapy context. Great efforts are currently directed toward the development of targeted therapies which often combine chemotherapeutics/immunotherapeutics with unprecedented nanomaterials [35,36]. Notwithstanding the great impact of the nanotechnology revolution in the field of drug delivery, with the commercialization and clinical application of some traditional anticancer drugs in the form of NPs [37,38], an increase in the selectivity of nanosystem uptake by the target organs is still necessary. To this end, we coupled the peptides with GNPs obtained by laser ablation in liquid (LAL) [39]. These GNPs are free of capping agents or ligands, and hence their surface is readily available for conjugation with thiolated compounds without the need for place-exchange reactions [40,41,42]. In addition, the GNPs are ultrapure because they are produced in aqueous solutions containing only 0.2 mM NaCl, without the addition of other chemical compounds which may interfere with the peptides structure and integrity [43]. For these reasons, and for the other advantages such as the fast, easy, reproducible, robust, clean, green and cost-effective synthesis, GNPs obtained by LAL have been frequently exploited as building blocks in bioconjugates for biotechnological applications [40,41,42,43,44].

## 2. Results

### 2.1. Peptide Synthesis

Trichogin analogs were produced by manual solid phase peptide synthesis (SPPS) following the procedure described in [21]. The peptide sequences are reported in Table 1, together with the plant-protection properties of the known peptides. The protocol foresees the use of ecofriendly reagents, such as Oxyma pure [45] and diisopropylcarbodiimide (DIC) as active agents and the use of ecofriendly ethyl acetate/dimethylsulfoxide mixtures instead of dimethylformamide (DMF) as solvent [46,47], thus limiting the impact of the peptide synthesis on the environment. The good purity of the crude peptides allowed us to purify them to >95% by medium-pressure liquid chromatography using an Isolera Prime instrument (Biotage, Uppsala, Sweden), further reducing the waste of organic solvents compared to preparative HPLC purification.

Phytosanitary trichogin analogs endowed with positive charge(s) were included in the present work based on promising literature data on the cytotoxicity of some of them (Table 2). Cationic [48], membrane-active peptides should indeed have an advantage in interacting with tumor cells since the latter have a higher concentration of negatively charged glycoproteins and phosphatidylserine in their membrane than healthy cells [49,50]. The newly designed and produced sequences Leu4-NH2 and Api8-NH2 (Table 1) are cheaper versions of the related Leu4-Lol and Api8-Lol peptaibols (endowed with a C-terminal Leucinol instead of the -Leu-NH2) in that the SPPS 2-chlorotrityl resin preloaded with the 1,2-aminoalcohol Lol [51,52] is very expensive compared to the rink amide resin herein used to produce Leu4-NH2 and Api8-NH2. Leu4-NH2 was selected because of the peculiar biological activity of the trichogin analog Leu4-Lol [27], which was proven to be able to interact with phospholipid membranes while being inactive both towards bacteria and eukaryotic cells [27,29]. Such behavior makes it a promising candidate as a drug carrier, for example, bound to nanoparticles. To this aim, the related Leu4-SH (Table 1) bearing a lipoyl moiety to be anchored to nanoparticles [53] was produced. Api8-NH2 was chosen because its Lol-bearing counterpart Api8-Lol was found to be selective against cancer cell lines, leaving healthy eukaryotic cells unaffected (Table 2) [29]. Similarly, Api8-SH (Table 1) was produced to decorate gold nanoparticles and to study its potential as a drug carrier. To study cell internalization by flow cytometry of the two sequences Leu4 and Api8, the related fluorescein 5(6)-isothiocyanate (FITC)-labeled analogs (Leu4-FITC and Api8-FITC) were also synthesized on a 2-chlorotrityl resin preloaded with the ethylendiamine linker. The fluorescent dye was coupled at the C-terminus in solution, using Merck-supplied FITC. The new sequences designed and synthesized in the present work (Table 1) were all obtained in high yield (>60%) and purity (≥95%).

The chemical characterization of the new sequences by means of HPLC and high-resolution electrospray ionization mass analysis (HR-ESIMS) is reported in the Appendix A.

### 2.2. Synthesis and Characterization of Peptide-Decorated Gold Nanoparticles

GNPs were synthesized by LAL, according to a previously described procedure [39,42,43] in distilled water with 2 × 10^−4^ M NaCl. The final GNP concentration was 22.8 × 10^−9^ M (value obtained by UV absorption analysis using the protocol described in [54]), and the Feret average size was 13 nm (evaluated by dynamic light scattering, DLS). The concentration of thiol-bearing peptide analogs to be used to saturate the GNP surface was estimated from the nanoparticle concentration and size and assuming a packing density of two peptide molecules per square nanometer of nanoparticles. The final concentration of Api8-SH in the GNP-containing solution was 24.2 µM. A lower concentration of 18.2 µM was used for the hydrophobic peptide Leu4-SH to avoid GNP aggregation and precipitation. Indeed, the DLS spectra of peptide-decorated GNPs (Appendix A) showed that their mean radius is increased compared to naked GNPs, indicating the onset of GNP aggregation, probably due to peptide–peptide interactions. Api8-S-GNPs form aggregates with an average diameter of 44.13 nm, while Leu4-S-GNPs have a larger mean diameter (over 1000 nm). The absorption spectra of the GNPs before and after 5-hour incubation with the peptides (Appendix A) indicate the red shift (from 520 nm to 535 nm for Api8-SH and 550 nm for Leu4-SH) and the broadening of the plasmon absorption band, which are indicative of particle aggregation [55]. Transmission electron microscopy (TEM) images (Appendix A) further confirmed the different aggregation state of the GNP constructs, in agreement with DLS-derived mean diameters, since the free GNPs are homogeneously dispersed as isolated particles on the carbon film of the TEM grid, while peptide-decorated GNPs show different aggregates of particles.

### 2.3. Conformational Study by Circular Dichroism (CD)

A conformational study on the new peptides Api8-NH2 and Leu4-NH2 and related analogs was performed by CD [56]. The presence of a C-terminal amide allows the onset of an additional hydrogen bond compared with the corresponding 1,2-aminoalcohol, thus increasing the strength of the helical conformation, which is known to be essential for the biological activity of peptaibols [57]. Api is a C^α^-tetrasubstituted residue that shares the same helix-inducing ability of Aib [58,59]; therefore, Api8-NH2 is expected to maintain the helical structure of the parent peptide trichogin GA IV [60]. Leu4-NH2 should retain the peculiar helix–loop–helix conformation of the related Leu4-Lol (with a C-terminal Leucinol instead of the -Leu-NH2) [27]. Such a 3D-structure seems mainly responsible for its peculiar bioactivity [27]. Initially, the 3D-structure adopted by Api8-NH2, Leu4-NH2, Api8-FITC, Leu4-FITC, Api8-SH, and Leu4-SH have been characterized in solution by CD under a variety of experimental conditions (see Appendix A). The results highlighted the onset of a right-handed, mixed α-/3_10_-helical conformation for all the analogs [61,62], as expected, with a switch towards a full α-helix in the presence of micelles of sodium dodecyl sulfate in water, namely the membrane-mimicking environment tested. The CD profile of Api8-NH2 and Leu4-NH2 was also acquired in the presence of cells (cell line MCF-10A) (Figure 1). This study also provided information on the stability of the peptides in the biological environment [63]. Trichogin analogs are usually resistant to the action of proteolytic enzymes thanks to the presence of the noncoded α-amino acid Aib in their sequences [27,64]. Indeed, the CD analysis confirmed both the stability of the peptides in the presence of cells and the conservation of their helical structure, with the positions of the two negative maxima falling at the canonical wavelengths for a α-helix, namely about 208 and 220 nm [65]. Comparing the CD profiles of the peptides in water (Appendix A) and in the presence of cells (Figure 1) it is clear that the negative maximum centered at about 222 nm is more pronounced in the latter conditions for both peptides. This modification on CD spectra is usually associated with a switch towards a more α-helical conformation [61,62]. The profile in the presence of cells resembles that recorded in the membrane-mimicking environment (micelles of sodium dodecyl sulfate, SDS 100 mM in H_2_O; Appendix A) for both peptides. This observation seems to point out the presence of peptide–membrane interactions for both trichogin analogs also when in contact with cells.

The CD analysis was also performed on the peptides Api8-SH and Leu4-SH once linked to GNPs (Api8-GNPs and Leu4-GNPs) to confirm the persistence of the helical conformation (Figure 2). The analysis revealed a right-handed, mixed α-/3_10_-helical conformation for both peptides also when linked to the GNPs, in line with the results obtained for the corresponding free peptides.

### 2.4. Membrane Leakage Study

We verified the ability of Api8-NH2 and Leu4-NH2 to cause leakage of the entrapped 5 (6)-carboxyfluorescein (CF) dye from small unilamellar vesicles (SUVs) [66] made of two lipid mixtures: phosphatidylethanolamine (PE)/phosphatidylglycerol (PG) 7:3, namely a phospholipid composition with a net negative charge, thus mimicking the tumor cell membrane [50]; and phosphatidylcholine (PC) and cholesterol (Ch) 7/3, a zwitterionic model membrane mimicking the erythrocytic one [67]. The assay follows fluorescence increase in response to CF release from SUVs in terms of percentage with respect to the full membrane disruption caused by Triton addition. The results are reported in Figure 3.

As expected, the peptides are able to cause membrane leakage with subsequent release of the fluorescent dye almost as effectively as the parent peptide trichogin GA IV. The results are in line with the literature on Leu4-Lol [27]. This result allowed us to conclude that both peptides retain the ability to interact with model membranes of the parent peptide. Leakage experiments cannot be performed on FITC-containing peptides since fluorescence from FITC interferes with that from the CF release.

### 2.5. In Vitro Cytotoxicity of Trichogin GA IV Analogs in Breast Cells

The in vitro anticancer activity of trichogin analogs with plant-protection properties (Table 1) was assessed by comparing the exerted cytotoxicity in MDA-MB-231 TNBC cells and in normal epithelial breast MCF-10A cells. Table 2 summarizes the cytotoxic activity against other cancer cell lines previously reported in the literature for some of them.

**Table 2 ijms-24-05537-t002:** Overview of the cytotoxic activity reported in the literature for some of the phytopharmaceutical peptides herein studied, together with selected analogs thereof as reference compounds.

Acronym	IC_50_ ^a^ (µM) and Target Cancer Cell Lines ^b^					Hemolysis/ Healthy Cells	References
	HeLa	A549	A431	T67	Ov-HL		
TRIC	4–8	4–6	4–6	2	n.d. ^c^	nonhemolytic	[29,30]
Api8-NH2	- ^d^						this work
*Api8-Lol * ^e^	>20 ^f^	n.d.	n.d.	13	n.d.	nonhemolytic ^f^/ nontoxic ^g^	[29]
Leu4-NH2	*-*						this work
*Leu4-Lol * ^e^	>20	nt	nt	>20	n.d.	nonhemolytic/ nontoxic	[29,68]
K2569-Lol	*-*						this work
K259-NH2	*-*						this work
K259-Lol	6	8	5	n.d.	n.d.	nonhemolytic	[68]
K25-Lol	7	10	5	n.d.	7–13	nonhemolytic <16 µM	[68]
K56-Lol	10	16	8	7	n.d.	nonhemolytic/ nontoxic	[29,68]
K6-NH2	n.d.	n.d.	n.d.	n.d.	7–13	n.d.	[28]
*K6-Lol * ^e^	2–8	8	5	4	n.d.	nonhemolytic <16 µM	[28,29,30]
K2-NH2	*-*						this work
*K2-Lol * ^e^	7	8	6	n.d.	n.d.	nonhemolytic	[30,68]
Api8-FITC	*-*						this work
Leu4-FITC	*-*						this work
Api8-SH	*-*						this work
Leu4-SH	*-*						this work

^a^ Compound concentration causing 50% of cell death by LDH (lactate dehydrogenase release) assay; ^b^ A549, lung cancer; A431, epidermoid cancer; T67, human glioma; Ov-HL, ovarian cancer and Hodgkin lymphoma; ^c^ not determined; ^d^ cytotoxicity never evaluated before; ^e^ Selected analogs differing from the new ones herein studied by the presence of a C-terminal leucinol, Lol: reported in italics, as reference; ^f^ nonhemolytic up to 20 µM; ^g^ nontoxic up to 20 µM.

Cells were incubated with increasing concentrations of peptides for 24 h before assessing cell viability with MTS assay in order to create dose/response curves (Figure 4) and calculate the half maximal inhibitory concentration (IC_50_) values for each peptaibol (Table 3). For convenience, peptides have been subdivided based on their C-terminal substituents, i.e., leucinol (lol) or amide (-NH_2_).

Leu4-NH2, Api8-NH2 and Leu4-Lol did not significantly affect the viability of both cell lines in the peptide concentration range tested (Figure 4). For all the other peptaibols, a concentration-dependent cell viability reduction was observed. The cytotoxicity is higher against the cancer than the normal cell line, as reflected by the calculated IC_50_ values (Table 3). Indeed, while the IC_50_ values measured in MDA-MB-231 cells are in the concentration range 8–12 µM (with the exception of K259-NH2, with an IC_50_ of about 13 µM), viability in MCF-10A cells did not drop below 50% for any of the peptides tested in the aforementioned concentration range. Of note, the IC_50_ values on normal cells are 5 µM higher than those measured in cancer cells. We note that none of the peptaibols were found cytotoxic against epithelial breast cells up to 10 µM, while in cancer cells most of them exerted significant viability reduction (killing activity of about 90% for K6-NH2 and K2-NH2, 60% for K259-Lol and K25-Lol, 30% for K2569-Lol, 20% for K56-Lol). This result is in line with the observation that K6-NH2 is taken up faster by ovarian cancer (OvCa) and Hodgkin lymphoma (HL) cell lines than normal cells [28]. Generally, the C-terminal moiety does not seem to significantly impact on peptide cytotoxicity, the effect of -NH2 or -Lol analogs being quite comparable. Again, this observation confirms and broadens the significance of the data reported for K6-Lol and K6-NH2 on OvCa and HL cell lines [28]. The cost-effective amino alcohol-to-amide substitution at the peptide C-terminus can therefore be considered conservative in terms of cytotoxic activity.

### 2.6. In Vitro Uptake of Selected Peptaibols

For this study, we decided to focus our attention on the unique, noncytotoxic, yet membrane-active peptides Leu4-NH2 and Api8-NH2 to study them as potential cancer-targeting agents. To assess if they displayed any selectivity for breast cancer vs. breast normal cells, we synthesized their FITC-conjugated version, Api8-FITC and Leu4-FITC, and studied their internalization by measuring the cellular uptake by flow cytometry (Figure 5). As reference, K6-NH2 internalization was measured as well, since for this specific peptide, previous studies already reported an increased uptake in tumor cells (HDLM-2 and A2780 tumor cell lines) with respect to normal healthy cells, such as peripheral blood mononuclear cells (PBMCs) or fibroblasts [28].

As clearly visible in Figure 5, the internalization rate of all the three peptide was significantly higher in MDA-MB-231 cells with respect to MCF-10A cells, indicating some cancer cell selectivity. Moreover, while the increase in concentration did not result in an increased uptake in normal cells, a concentration-dependent uptake was measured for all three peptides in cancer cells. At the highest concentration tested, Leu4-FITC, Api8-FITC and K6-FITC were internalized about 5, 3, and 4-folds more, respectively, in MDA-MB-231 with respect to MCF-10A cells. Of note, none of the FITC-conjugated peptaibols were cytotoxic in both cell lines up to 10 µM (Appendix A).

### 2.7. Uptake of GNPs Decorated with Peptaibols in Breast Cancer Cells and Macrophages

Cell uptake of the peptide-decorated GNPs was assessed after 24 h incubation by transmission electron microscopy (TEM) in samples counterstained with uranyl acetate. The results were further confirmed by atomic absorption spectrometry (AAS). First, we verified the compatibility of the peptide-decorated GNPs toward our in vitro cell models, in order to exclude any increased uptake due to their possible intrinsic cytotoxicity. As confirmed by the graphs of Appendix A, peptide-decorated GNP suspensions were almost noncytotoxic against both cell lines after 24 h cell incubation up to a concentration corresponding to 10 µM of peptide (with the exception of a few cases in which we measured a 15% viability reduction; see statistic for significance). A concentration of peptide-GNPs corresponding to 5 µM peptide, devoid of any possible cytotoxic effect, was thus used for TEM analysis (Figure 6).

In Figure 6a,b are reported a few representative images, showing that all GNP samples are internalized in both cell lines but to a significantly different extent. Indeed, completely different trends of GNP uptake were observed both among cell lines and among the samples. Unexpectedly, by observing tens of TEM-acquired images, it appeared that all types of GNPs were taken up to a greater extent in normal MCF-10A cells than in the tumor cells, in the increasing order: Leu4-GNPs < Api8-GNPs < GNPs. The opposite trend was observed in tumor MDA-MB-231 cells, with Leu4-GNPs showing the highest intracellular uptake, followed by Api8-GNPs and nude GNPs. Therefore, it appears clear that: (i) GNP functionalization impacts the internalization rate; (ii) GNP uptake is cell-line dependent; (iii) GNPs are internalized by endocytosis (Figure 6). Indeed, MDA-MB-231 invariably contained groups of GNPs entrapped in vesicular components in the cytoplasm (e.g., endosome/lysosomes); magnified images 5, 8, and 11 (Figure 6a) clearly show that those organelles contain a higher number of NPs in the case of Leu4-GNPs, followed by Api8-GNPs, and nude NPs. In MCF-10A cells, the elevated number of GNPs internalized, especially in the case of nude GNPs and Api8-GNPs, entailed the formation of larger vesicles, with hundreds of NPs inside (Figure 6(b6,b9)). To verify TEM internalization results with a more quantitative analysis, we performed AAS under the same experimental conditions. AAS measurements (Figure 6c) indeed confirmed the internalization trend previously observed: the uptake of all GNP formulations was significantly higher in MCF-10A than MDA-MB-231 cells, with a ratio of about 5.5: 3.5: 1.2 fold for nude GNPs, Api8-GNPs, Leu4NPs, respectively. Importantly, AAS data revealed that the presence of peptaibols on the surface of GNPs increased GNP accumulation in TNBC cells (with respect to the nude counterpart) while decreasing it in normal epithelial cells of the same derivatization. Of note, using Leu4 as targeting agent resulted in a quite similar internalization rate among our two model cell lines. DLS measurements on Leu4-GNPs indicated an average hydrodynamic diameter > 1000 nm, very likely due to the formation of aggregates between particles in those experimental conditions. On the other hand, as shown in Appendix A, the diameter of single Leu4-S-GNPs does not exceed 30–40 nm, i.e., the same dimension range as that of Api8-S-GNPs. Image 12 of Figure 6 (MDA-MB-231 cells) shows aggregated Leu4-GNPs interacting with plasma membrane as well as NPs within endocytic vesicles, confirming that, at least in the cell lines considered in this study, aggregates of gold NPs are promptly internalized. In any case, further experiments using other normal and cancer cell lines are needed to confirm the observations reported herein, which indicated Leu4 as the most appropriated peptaibol for targeted drug delivery or therapeutic applications. In the latter context, another fundamental characteristic that a nanosystem must possess is the capacity to avoid recognition/clearance by the reticuloendothelial system (RES) components, mainly macrophages (e.g., Kupffer cells) resident in liver and spleen [69]. NP recognition is mainly mediated by serum proteins absorbed on the surface of the nanosystem, the so called “protein corona”. Since the conjugation of GNPs with peptaibols could influence macrophage phagocytosis, we analyzed the interactions of the different GNP formulations with human macrophages, derived from monocytes isolated from buffy coats, by means of TEM analysis. As visible in the representative images of Appendix A, a very similar extent of GNP capture by macrophages was observed, independently of the presence of peptaibol decoration on NP surfaces.

## 3. Discussion

Trichogin is a naturally-occurring, cytotoxic peptaibol produced by a fungus of the genus *Trichoderma*, which is used in organic farming. Antitumor activity of several trichogin analogs endowed with plant protection properties have been assessed against breast cancer cells (cell line MDA-MB-231) as well as new analogs, obtained in high yield and purity via a protocol with limited environmental impact. The cytotoxicity to normal cells of the same derivation (cell line MCF-10A) was also tested. All Lys-containing trichogin analogs showed antitumor activity with an IC_50_ between 8 and 13 µM, a concentration range where healthy cells are still about 90% viable. Such Aib-containing, membrane-active cationic peptides represent a promising class of potential antitumor compounds since they combine affinity to tumor cell membranes—rich in phosphoserine [70] with a limited chance to elicit resistance since they do not have a specific target [71]. In addition, Aib is known to increase peptide proteolytic resistance [72,73]. The assays allowed us to identify two analogs (Leu4-NH2 and Api8-NH2) as potential targeting agents/drug carriers [74] since they displayed membrane activity devoid of cytotoxicity. Circular dichroism analysis carried out in the presence of breast cells showed the presence of the helical structure, thus proving the stability of the two analogs in the biological environment. GNPs decorated with the thiol-containing versions of those analogs were produced and characterized. Circular dichroism (CD) confirmed the presence of the helical conformation for the peptides even when anchored to the nanoparticles. Intracellular uptake was evaluated both in cancer and in the corresponding normal cell line. Peptide decoration increased GNP-uptake by cancer cells with Leu4-S-GNPs being internalized best. In the meantime, peptide-decoration also limited GNP-intake by normal cells, again with Leu4-S-GNPs showing the highest uptake reduction. NPs are known to be recognized by macrophages, thus limiting their possible side effects on off-target cells [75]. Peptide-decoration might perturb this feature [76]. Nonetheless, we found that GNP decoration with the two peptides did not alter GNP capture by macrophages.

## 4. Materials and Methods

### 4.1. Peptide Synthesis

Api8-NH2 and Leu4-NH2 peptides were obtained following the solid-phase peptide synthesis (SPPS) procedures described in [21,27] on a Rink Amide resin (Novabiochem, Merck Biosciences, La Jolla, CA, USA). The C-terminal FITC analogs Api-FITC and Leu-FITC were synthesized by manual SPPS on a 1,2-diaminoethane-trityl resin (Iris Biotech, Marktredwitz, Germany). Fmoc-deprotection was achieved by treatment with 20% piperidine solution in N,N-dimethylformamide (DMF). The deprotection step was repeated twice (5 and 10 min, respectively). The coupling steps were generally carried out exploiting Oxyme pure and diisopropylcarbodiimide (DIC) as activating agents. The coupling reactions involving Aib residues were doubled. All 1 h coupling steps were carried out with three equivalent excess of the activated residue. Capping of the N-terminal α-NH_2_ with 1-octanoyc acid was achieve by reaction with four equivalents of 1-Octanoic acid, Oxyma pure, and DIC. The 1 h coupling was repeated twice. Peptide cleavage from the Rink amide resin was achieved by 2.5 h acid treatment with the standard mixture: trifluoroacetic acid, TFA, 95%; water, 2.5%; triisopropylsilane, TIS, 2.5%. C-terminal amino alcohol-containing peptides were cleaved by several treatments with 30% hexafluoroisopropanol (HFIP) in dichloromethane, as described in [77]. The filtrates, after precipitation in diethyl ether, were collected and concentrated under a flow of N_2_. The crude peptides were used without purification to obtain the respective, FITC-bearing, compounds, by reaction with 2 equivalents of FITC in DMF in the presence of diisopropylethylamine. The workup of the reaction mixtures, dissolved in dichloromethane, involved washings with H_2_O. Each organic phase was washed with H_2_O three times, dried over Na_2_SO_4_, filtered, and evaporated to dryness. Boc removal from the precursor Api (Boc)-FITC was achieved by dissolving the peptide in HCl 3M in methanol. The reaction was left stirring until quantitative Boc-removal (2 h, followed by HPLC). Crude peptides were purified by medium-pressure liquid chromatography on a Biotage Isolera Prime Instrument. Yield: Api8-NH2, 60.28%; Leu4-NH2 72.5%. The purified fractions were characterized by analytical RP-HPLC on a Jupiter Phenomenex (Torrance, CA, USA) C_18_ column (4.6 × 250 mm, 5μm, 300 Å) using an Agilent (Santa Clara, CA, USA) 1200 HPLC pump. The binary elution system used was A, H_2_O/CH_3_CN (9:1 *v*/*v*) + 0.05% trifluoroacetic acid (TFA); B, CH_3_CN/H_2_O (9:1 *v*/*v*) + 0.05%TFA; flow rate 1 ml/min; spectrophotometric detection at λ = 216 nm. Electrospray ionization, high-resolution mass spectrometry (ESI-HRMS) was performed by using a Waters Micromass instrument (Milford, MA, USA). HPLC and ESI-MS spectra, and characterization of the synthetic segments obtained during solution phase synthesis are reported in the Appendix A.

### 4.2. Circular Dichroism

The CD spectra were obtained on a Jasco (Tokyo, Japan) J-1500 spectropolarimeter. Fused quartz cells (Hellma, Mühlheim, Germany) of 1 mm path length were used. The values are expressed in terms of [θ]_T_, total molar ellipticity (deg∙cm^2^∙dmol^−1^). Spectrograde MeOH and TFE (Acros, Geel, Belgium) were used as solvents. The CD measurements in cells were performed as follows: 1 × 10^6^ cells were harvested, washed twice in PBS and incubated in the same buffer with peptides (concentrations: Api8, 5 × 10^−5^ M; Leu4, 1 × 10^−5^ M) for 1 h at 37°C in a rotating mixer before recording the CD spectra.

### 4.3. Gold Nanoparticles Synthesis and Characterization

GNPs LAL synthesis was performed with the laser pulses at 1064 nm (6 ns, 50 Hz) focused to 8 J/cm^2^ with an f 100 mm lens on a 99.99% pure Au plate dipped in distilled water with 2 × 10^−4^ M NaCl [39,42,43]. The laser beam ablated a circular area of 5 mm in diameter and the ablation cell was mounted on a motorized XY scanning stage (Standa, Vilnius, Lithuania) managed with a 2-axis stepper and a DC motor controller. UV-vis spectroscopy was performed in 0.2 cm optical path quartz cuvettes using a JASCO (Tokyo, Japan) V770 spectrophotometer. DLS analysis was performed using a Malvern (Malvern, UK) Zetasizer Nano ZS in ZEN0040 cells. TEM was performed on a FEI (Hillsboro, OR, USA) Tecnai G2 12 instrument operating at 100 kV and equipped with a Veleta (Olympus Soft Imaging System, Denver, CO, USA) digital camera.

Peptide decoration was performed by mixing a concentrated solution of peptide with the solution of GNPs as obtained from LAL, obtaining a final concentration of 24.2 µM for Api8-SH and 18.2 µM for Leu4-SH, respectively. The solution was incubated overnight, then washed to remove excess peptide before use.

### 4.4. Leakage

Phospholipids (phosphatidylethanolamine (PE) and phosphatidylglycerol (PG)) were supplied by Avanti Polar Lipids (Alabaster, AL, USA). The two lipids were combined in a 7:3 *w*/*w* ratio in a test tube, dissolved in chloroform, and then dried over a nitrogen flux to obtain a lipid film. The lipid film was then hydrated with a solution of 5(6)-carboxyfluorescein (CF) in 30 mM HEPES buffer (pH 7.4) at room temperature overnight. The resulting vesicle suspension was sonicated twice for 30 min over an ice bath to break the multilamellar structure, to obtain small unilamellar vesicles (SUVs). The suspension was loaded in a gel filtration column Sephadex G-75 (Merck) to remove the excess of CF. The obtained SUVs concentrated solution was diluted with buffer (100 mM NaCl, 5 mM HEPES, pH 7.4) to a working concentration of 0.06 mM. The working solution of SUVs was stored at 4 °C and used within 24 h. Fluorescence (measured on a MPF-66 spectrofluorimeter, Perkin Elmer, Waltham, MA, USA) was used to evaluate peptide-induced dye leakage from the liposomes. Onto each cuvette containing a fixed volume of the SUVs working solution (2.5 mL), increasing amounts of peptide solution (s) in water (or methanol for the non-water-soluble analogs) were added to achieve increasing [peptide]/[lipid] molar ratios (R^−1^). The fluorescence was recorded for each cuvette, at 520 nm with λ_exc_ = 488 nm, before peptide addition and after 20 min incubation with the peptide. The released CF at time *t* (%CF) was determined as: %CF = (F_t_ − F_0_)/(F_T_ − F_0_) × 100, where F_0_ is the fluorescence intensity recorded for SUVs before peptide addition; F_t_ is the fluorescence intensity of vesicles at the time t: 20 min after peptide addition; and F_T_ is the total fluorescence intensity determined by disrupting the SUVs by adding 10% *v*/*v* Triton X-100 in water (50 μL). The results are reported on a logarithmic scale.

### 4.5. Cell Lines

MDA-MB-231 (human triple-negative breast cancer) and MCF10A (human nonmalignant breast epithelial) cell lines were purchased from American Type Culture Collection (ATCC, Rockville, MD, USA). MDA-MB-231 were grown in Dulbecco’s Modified Eagle Medium (DMEM) with Glutamax^TM^ supplemented with 10% heat inactivated fetal bovine serum (FBS) and antibiotics (100 U/mL streptomycin, 100 μg/mL penicillin G). MCF10A cells were cultured in DMEM/F12 medium supplemented with 5% horse serum, 20 ng/mL EGF, 0.5 mg/mL hydrocortisone, 100 ng/mL cholera toxin, 10 μg/mL insulin, and antibiotics.

Human macrophages were derived from monocytes extracted from fresh buffy coats of healthy donors (obtained from Azienda Ospedaliera Padova, Padova, Italy) by centrifugation over a Ficoll–Hypaque step gradient and subsequent Percoll gradient (Sigma-Aldrich, Munich, Germany).

All cell culture medium and supplements were purchased from Life Technologies or Sigma-Aldrich (Munich, Germany), while sterile plasticware was purchased from Falcon^®^ (Corning, New York, NY, USA).

### 4.6. In Vitro Cytotoxicity of Peptaibols and Peptaibol-Decorated GNPs

The cytotoxicity of peptides and free Api8- and Leu4-decorated GNPs was assessed with the MTS assay (CellTiter 96^®^ AQueous One Solution Cell Proliferation Assay, Promega, Milan, Italy) in cancer MDA-MB-231 and normal MCF-10A cells exposed to increasing concentrations of peptaibols/GNPs for 24 h. Cells (8 × 10^3^ cells/well for MDA-MB-231, 6 × 10^3^ cells/well for MCF-10A) were seeded in 96-well plates, and after 24 h the medium was replaced with a fresh one containing peptaibols/GNPs. For MTS assay, the medium was replaced with 100 μL of serum-free medium and 20 μL of the CellTiter 96^®^ reagent. After 60–90 min, the absorbance at 492 nm was measured with a Multiskan Go (Thermo Fischer Scientific, Carlsbad, CA, USA) plate reader and cell viability was expressed as a function of absorbance relative to that of control cells (considered as 100% viability). IC_50_ values for each peptaibol treatment were extrapolated from the relative dose–response curves obtained with the software GraphPad Prism 9.5.

### 4.7. Cellular Uptake of FITC-Conjugated Peptides

Flow cytometry and peptides conjugated with FITC, namely Api8-FITC and Leu4-FITC, were used to assess the capacity of peptaibols to be internalized by breast cancer cells, Briefly, 8 × 10^4^ MDA-MB-231 and MCF-10A cells/well were grown in 24-well plates for 24 h and then incubated for 24 h with 5 μM peptaibols. At the end of the incubation time, cells were washed twice with Versene solution and detached from the plates with trypsin that was neutralized by the addition of FBS. Those washings both before cell detachment from plates and after cell recovery and centrifugation removed any peptides not strongly associated with cell plasma membranes, assuring FITC signals measured by flow cytometry were referred exclusively to internalized peptides. Cells were centrifuged and resuspended in Versene before measuring FITC fluorescence using a BD Fortessa^TM^ X-20 flow cytometer (Becton Dickinson, San Jose, CA, USA). For each sample, 10^4^ events were acquired and analyzed using the FACSDiva 9.0 software.

### 4.8. Cellular Uptake of GNPs Measured by TEM

The uptake of GNPs and peptide-decorated GNPs by breast cells was assessed by TEM. Briefly, 8 × 10^4^ MDA-MB-231 and MCF-10A cells/well were grown in 24-well plates for 24 h and then incubated for 24 h with a GNPs concentration corresponding to a final 5 μM of peptide. Cells were then fixed in 2.5% glutaraldehyde in 0.1 M phosphate buffer at pH 7.4 for 1 h at room temperature and then washed three times with phosphate buffer (10 min each wash). The samples were post-fixed in 1% osmium tetroxide in 0.1 M phosphate buffer pH 7.4 for 1 h at room temperature and dehydrated in ethanol from 10 to 100% (three times) for 10 min each step and then included in epoxy resin. The samples were sectioned with an Ultrotome V ultramicrotome (LKB instruments, Victoria, TX, USA). Thin sections (80–100 nm) were counterstained with uranyl acetate and lead citrate and then observed with a Tecnai G2 transmission electron microscope (FEI Company, Hillsboro, OR, USA) operating at 100 kV.

TEM analysis was performed also to assess the extent of recognition and capture by human macrophages. Briefly, 3 × 10^5^ monocytes isolated from buffy coats were seeded per well in 24 wells/plate and cultured for 7 days in RPMI-1640 medium (Life Technologies, Delhi, India) supplemented with 20% FBS and 100 ng/mL of human macrophage colony-stimulating factor (Peprotech, Neuilly-sur-Seine, France) to promote macrophage differentiation. On day 4 from the seed, the macrophage colony-stimulating factor was added again. For AuNP uptake experiments, 7-day-old macrophages were incubated for 24 h with GNPs, Leu4-GNPs, or Api8-GNPs with a final peptide concentration of 5 μM. The samples were then processed as mentioned above for mammary cell lines.

### 4.9. Cellular Uptake of GNPs Measured by Atomic Absorption Spectroscopy

Briefly, 4 × 10^5^ MDA-MB-231 and MCF-10A cells/well were seeded in 12-well plates and allowed to growth for 24 h before incubating them with GNPs or peptide-decorated GNPs (at a 5 μM peptide concentration). At the end of the 24 h GNP incubation, cells were washed twice with PBS and detached from the plates with trypsin that was neutralized by the addition of FBS. Cells were centrifuged and washed twice with Versene solution. A fraction of cell suspension was taken for subsequent quantification of total protein content using Pierce BCA Protein Assay kit (Thermo Fisher, Waltham, MA, USA), while the remaining suspension was digested using a mixture consisting of 1 part of 33% hydrogen peroxide and 2 parts of 27% nitric acid. The obtained solutions were then diluted 1:100 using purified water and analyzed by atomic absorption spectroscopy (AAS) to assess gold concentration on a Varian AA240 Zeeman instrument equipped with a GTA120 graphite furnace, a Zeeman background corrector, and an autosampler (Varian Inc., Palo Alto, CA, USA). The total gold content was then expressed as $\frac{\mathrm{ng}\;\mathrm{Au}}{\mu g\;\mathrm{protein}}$.

## 5. Conclusions and Future Perspectives

The present article described the promising antitumor properties of structural analogs of the natural peptide trichogin GA IV and highlighted the chance to exploit them also as targeting agents by exploiting the unique combination of effective membrane selectivity and lack of cytotoxicity. We demonstrated Leu4-NH2 and Api8-NH2 selectivity towards breast tumor cell, both alone and when conjugated to GNPs, thus opening the way to their functionalization with anticancer drugs. Future work will take advantage of the versatility of the two trichogin analogs, which can indeed be easily linked to chemotherapeutics (e.g., doxorubicin) through bioresponsive linkers able to promote drug release exclusively in the tumor microenvironment, thus further enhancing the overall therapeutic selectivity. (both alone and linked to GNPs) to known in the last decade, GNPs are receiving increasing attention for their useful application in sonodynamic therapy (SDT), an emerging and poor invasive anticancer therapeutic technique based on the targeted administration of ultrasounds to activate sensitizer molecules [77]. In this connection, our trichogin-analog-functionalized GNPs could represent a promising sensitizer carrier, endowed with cancer-cell selectivity, to be combined with ultrasounds.

## Figures and Tables

**Figure 1 ijms-24-05537-f001:**
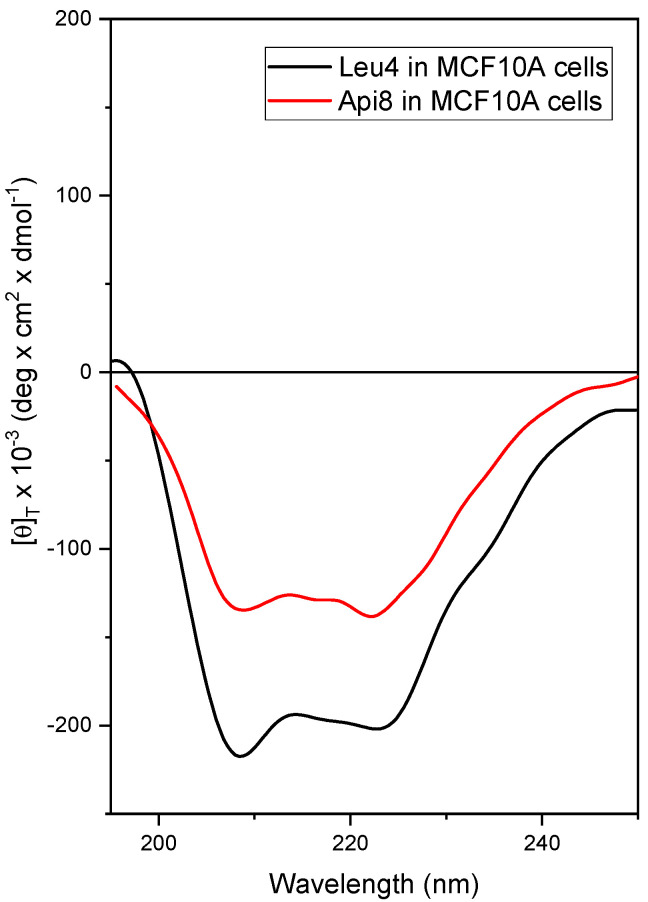
CD profiles of Leu4-NH2 and Api8-NH2 acquired at 25 °C in the presence of MCF-10A epithelial breast cells. (Peptide concentrations: Api8, 5 × 10^−5^ M; Leu4, 1 × 10^−5^ M).

**Figure 2 ijms-24-05537-f002:**
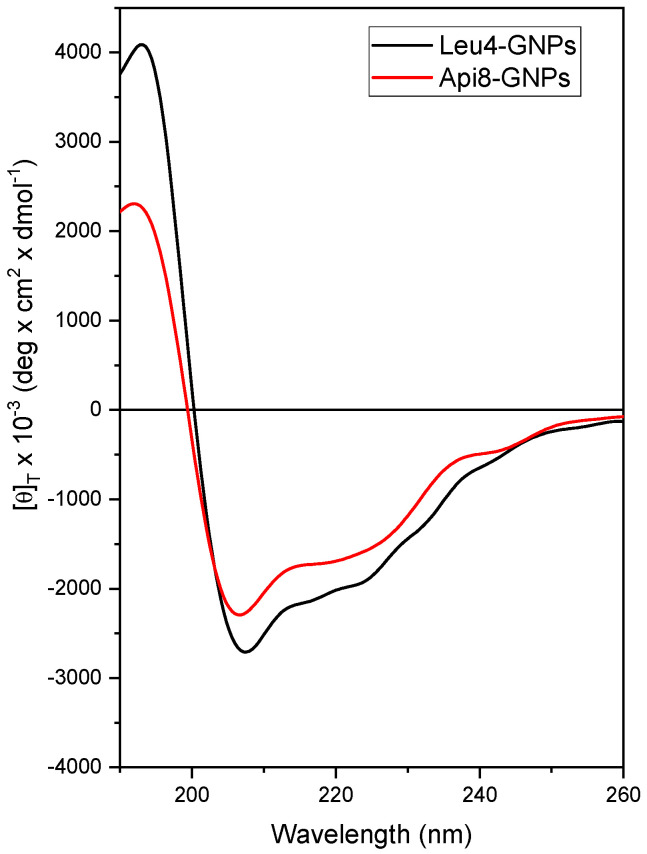
CD profiles of GNPs after decoration with Leu4-SH or Api8-SH, acquired at 25 °C in H_2_O (peptide-GNP concentration: 10^−6^ M).

**Figure 3 ijms-24-05537-f003:**
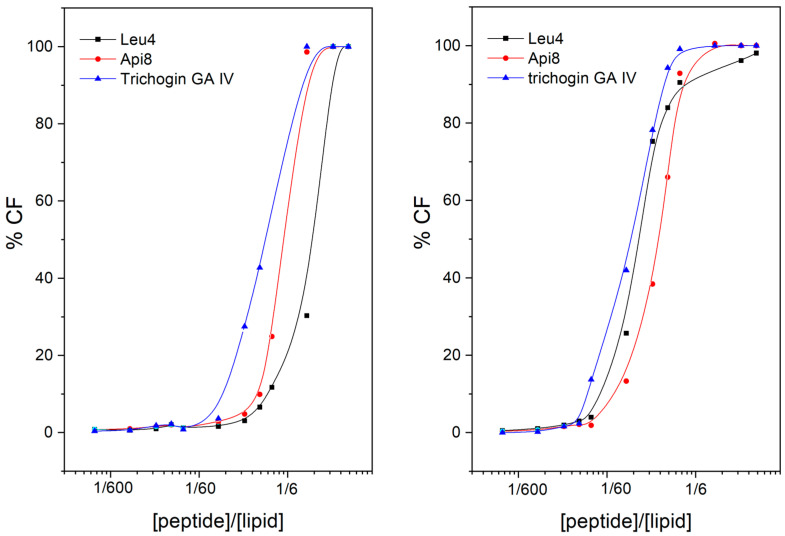
CF leakage from SUVs (**left**, PC/Ch 7:3; **right**, PE/PG 7:3. Lipid concentration, 60 μM) induced by increasing peptide concentration.

**Figure 4 ijms-24-05537-f004:**
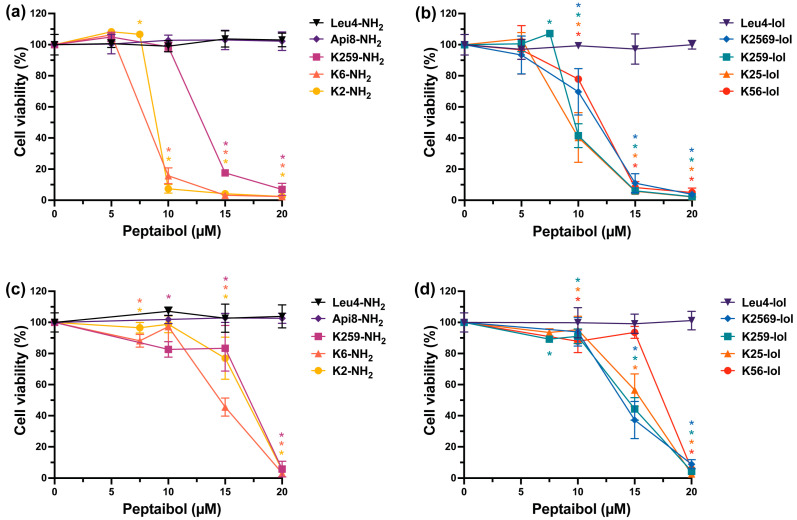
(Cytotoxicity of trichogin GA IV analogs toward triple negative breast cancer cells MDA-MB-231 (**a**,**b**) and normal breast cells MCF-10A (**c**,**d**). Cells were exposed to increasing concentrations of peptaibols for 24 h, and viability was measured with the MTS assay at the end of the incubation time. Data are expressed as mean percentage ± SD of at least three independent experiments, carried out in triplicate. *: *p* < 0.001, significantly different from controls (*t*-test).

**Figure 5 ijms-24-05537-f005:**
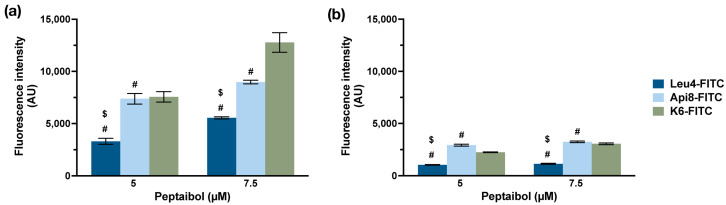
Flow cytometry measurements of FITC-conjugated peptaibols in MDA-MB-231 (**a**) and MCF-10 (**b**) cells after a 24 h incubation period. Data are expressed as mean percentage ± SD of at least two independent experiments, carried out in triplicate. #: *p* < 0.05, significantly different from K6-FITC, $: *p* < 0.05, significantly different from Api8-FITC (two-way ANOVA with Bonferroni′s correction).

**Figure 6 ijms-24-05537-f006:**
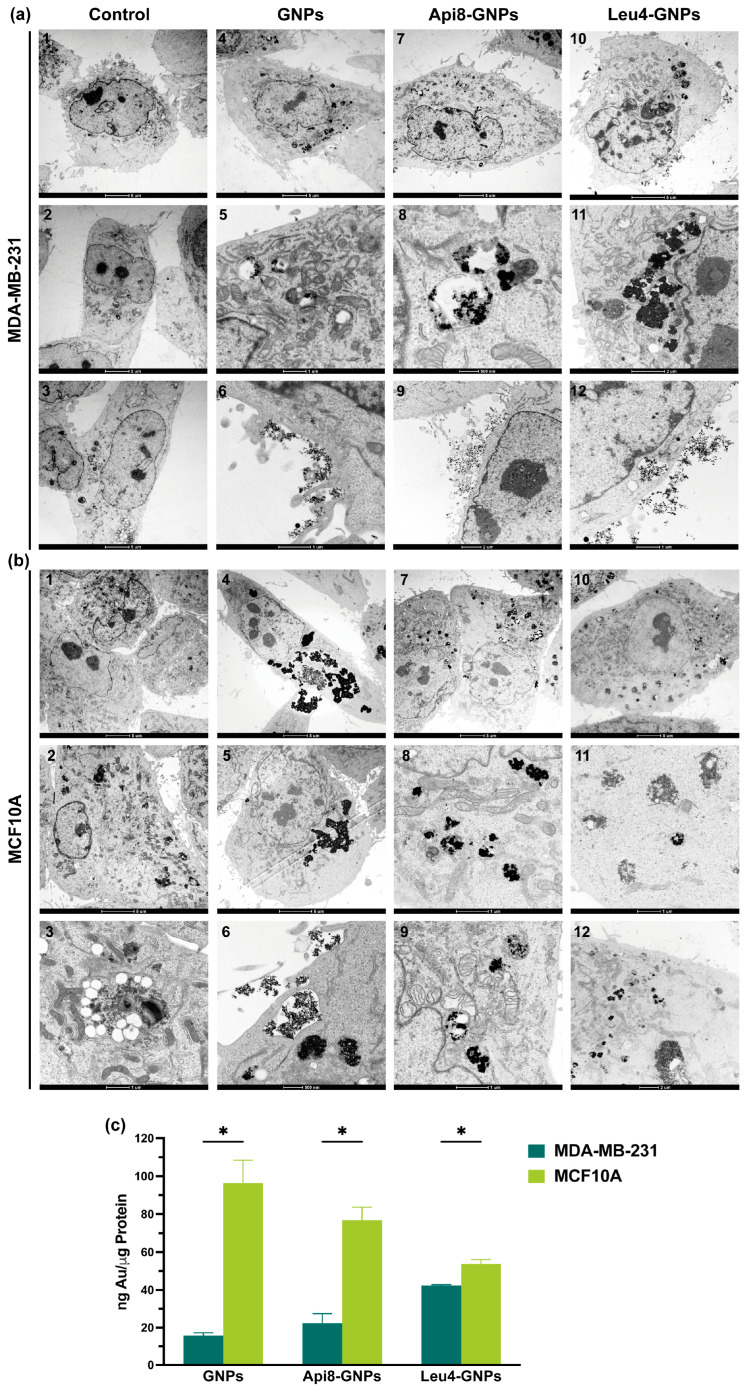
TEM images acquired at different magnifications of MDA-MB-231 (**a**) and MCF-10A (**b**) cells incubated for 24 h with free GNPs, Api8-GNPs, or Leu4-NPs. Scale bars: 5 µm, images (**a1**)–(**a4**), (**a7**), (**a10**), (**b1**), (**b2**), (**b4**), (**b5**), (**b7**), (**b10**); 2 µm, images (**a9**), (**a11**), (**b12**); 1 µm, images (**a5**), (**a6**), (**a12**), (**b3**), (**b8**), (**b9**), (**b11**); 500 nm, images (**a8**), (**b6**). (**c**) Uptake of the GNP formulations measured by atomic absorption spectrometry (AAS). The quantity of internalized GNPs inside samples is reported as ng of gold and expressed with respect to the μg of protein content. Data are expressed as mean percentage ± SD of at least two independent experiments, carried out in triplicate. *: *p* < 0.001, significantly different from MCF-10A cells (*t* test).

**Table 1 ijms-24-05537-t001:** Primary structure of the trichogin analogs studied in this work.

Acronym	Sequence ^a^
TRIC ^b^	Oct-Aib^1^-Gly-Leu-Aib^4^-Gly-Gly-Leu-Aib^8^-Gly-Ile-Lol^11^
Api8-NH2	Oct-Aib^1^-Gly-Leu-Aib^4^-Gly-Gly-Leu-**Api**^8^-Gly-Ile-Leu^11^-NH_2_
Leu4-NH2	Oct-Aib^1^-Gly-Leu-**Leu**^4^-Gly-Gly-Leu-Aib^8^-Gly-Ile-Leu^11^-NH_2_
K2569-Lol	Oct-Aib^1^-**Lys**-Leu-Aib^4^-**Lys**-**Lys**-Leu-Aib^8^-**Lys**-Ile-Lol^11^
K259-NH2	Oct-Aib^1^-**Lys**-Leu-Aib^4^-**Lys**-Gly-Leu-Aib^8^-**Lys**-Ile-Leu^11^-NH_2_
K259-Lol	Oct-Aib^1^-**Lys**-Leu-Aib^4^-**Lys**-Gly-Leu-Aib^8^-**Lys**-Ile-Lol^11^
K25-Lol	Oct-Aib^1^-**Lys**-Leu-Aib^4^-**Lys**-Gly-Leu-Aib^8^-Gly-Ile-Lol^11^
K56-Lol	Oct-Aib^1^-Gly-Leu-Aib^4^-**Lys**-**Lys**-Leu-Aib^8^-Gly-Ile-Lol^11^
K6-NH2	Oct-Aib^1^-Gly-Leu-Aib^4^-Gly-**Lys**-Leu-Aib^8^-Gly-Ile-Leu^11^-NH_2_
K2-NH2	Oct-Aib^1^-**Lys**-Leu-Aib^4^-Gly-Gly-Leu-Aib^8^-Gly-Ile-Leu^11^-NH_2_
Api8-FITC	Oct-Aib^1^-Gly-Leu-Aib^4^-Gly-Gly-Leu-**Api**^8^-Gly-Ile-Leu^11^-NH-(CH_2_)_2_-NH-**FITC**
Leu4-FITC	Oct-Aib^1^-Gly-Leu-**Leu**^4^-Gly-Gly-Leu-Aib^8^-Gly-Ile-Leu^11^-NH-(CH_2_)_2_-NH-**FITC**
Api8-SH	Oct-Aib^1^-Gly-Leu-Aib^4^-Gly-Gly-Leu-**Api**^8^-Gly-Ile-Leu^11^-NH-(CH_2_)_2_-NH-**Lipoyl**
Leu4-SH	Oct-Aib^1^-Gly-Leu-**Leu**^4^-Gly-Gly-Leu-Aib^8^-Gly-Ile-Leu^11^-NH-(CH_2_)_2_-NH-**Lipoyl**

^a^ Oct, *n*-octanoyl; Aib, α-aminoisobutyric acid; Lol, 1,2-aminoalcohol leucinol; FITC, fluorescein isothiocyanate; Lipoyl, acyl moiety from lipoic acid; Api, 4-aminopiperidin-4-carboxylic acid. ^b^ Native sequence of trichogin GA IV. Modifications from the native sequence are highlighted in bold.

**Table 3 ijms-24-05537-t003:** IC_50_ (expressed in µM) values measured in MDA-MB-231 and MCF-10A cells exposed to peptaibols for 24 h.

Acronym	MDA-MB-231	MCF-10A
Api8-NH2	>20	>20
Leu4-NH2	>20	>20
Leu4-Lol	>20	>20
K2569-Lol	12	14
K259-NH2	13	17
K259-Lol	10	15
K25-Lol	9	16
K56	12	18
K6-NH2	8	15
K2-NH2	9	17

## Data Availability

Data supporting reported results can be obtained from the authors.

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
