# Peer review of "Anticancer and Targeting Activity of Phytopharmaceutical Structural Analogs of a Natural Peptide from Trichoderma longibrachiatum and Related Peptide-Decorated Gold Nanoparticles"

_ijms, 2023, doi:10.3390/ijms24065537_

Round 1
Reviewer 1 Report
The paper by Moret et al. describes the synthesis and characterization of tricogin analogues.
In my opinion the title of the paper does not reflect the conclusion of the work.
The authors focused their attention on two peptides, Leu4-NH2 and Api8-NH2 and measured for these leackage from SUV.
Peptides Leu4-NH2 and Api8-NH2 are not cytotoxic ; their FITC derivatives are apparently internalized by cancer cell to a higher extent as compared to healthy cells (internalization vs absorption on the cell membrane should be assessed by confocal microscopy experiments). For this reason the two peptides were identified as breast cancer cell targeting peptides. Decoration of GNPs with the peptides causes an increase in their uptake by breast cancer cell.
Title of the work should reflect the cancer cell targeting activity of these peptides, rather then their anticancer activity.
The synthesis of peptides and their characterization by CD is clearly described.
The CD experiments of peptides with cells are very interesting. In which conditions were these experiments performed? (buffer, amount of cells…) For sake of clarity conditions should be reported. A comment on the comparison of the secondary structure of peptides with and without cells should be added. I don’t see in the CD of spectrum of Leu4-NH2 in buffer or in water. If not present, this should be added.
One important issue that needs to be clarified is the following: Leu4-GNP are very large, as assessed by DLS measurements. It seems quite strange that such big particles are uptaken by cells. Can the author explain this?
The relationship between the results of leackage experiments and the uptake should be better discussed. In addition it would be desirable to see how K6-FITC performs in membrane leackage and on GNP and also discuss uptake results in comparison with tricogin.
In Figure 3, please change “lipide” to “lipid”.
Finally, it is clear that a huge amount of work has been performed but presentation of results is in my opinion a little confusing. There are too many information in the tables.
As the topic of the paper is the anticancer/cancer cell targeting activity of TRIC analogue the information on PPP in table 1 is not useful. Table 2 could be moved to the Supplementary, as the activity of the peptides is not even reportd on the same cell line tested in this work.
Author Response
REVIEWER 1
Referee: The paper by Moret et al. describes the synthesis and characterization of tricogin analogues. In my opinion the title of the paper does not reflect the conclusion of the work. The authors focused their attention on two peptides, Leu4-NH2 and Api8-NH2 and measured for these the leakage from SUV. Peptides Leu4-NH2 and Api8-NH2 are not cytotoxic; their FITC derivatives are apparently internalized by cancer cell to a higher extent as compared to healthy cells (internalization vs absorption on the cell membrane should be assessed by confocal microscopy experiments). For this reason, the two peptides were identified as breast cancer cell targeting peptides. Decoration of GNPs with the peptides causes an increase in their uptake by breast cancer cell. Title of the work should reflect the cancer cell targeting activity of these peptides, rather than their anticancer activity.
Authors: We thank the referee for the very useful and constructive comments to our work, which allowed us to improve the manuscript. We modified the title of the paper as suggested. It now reads: “Anticancer and Targeting Activity of Phytopharmaceutical Structural Analogs of a Natural Peptide from Trichoderma longibrachiatum and Related Peptide-Decorated Gold Nanoparticles”.
R. FITC derivatives are apparently internalized by cancer cell to a higher extent as compared to healthy cells (internalization vs absorption on the cell membrane should be assessed by confocal microscopy experiments)
A. We thank the reviewer for this comment. We have validated our internalization results, assuring FITC signals measured by flow cytometry are referred exclusively to internalized peptides, as follows: before cell detachment from plates as well as after cell recovery and centrifugation, cell samples were washed twice with PBS to detach peptides only slightly associated with cell plasma membranes. Moreover, our TEM studies in cells clearly highlight the presence of the peptides in the intracellular space. We therefore decided not to perform additional tests, to not postpone the deadline for submission of the revision, kindly set by the Editor. We have added a description on results validation in the methods session related to the internalization.
R. The synthesis of peptides and their characterization by CD is clearly described. The CD experiments of peptides with cells are very interesting. In which conditions were these experiments performed? (buffer, amount of cells…) For sake of clarity conditions should be reported.
A. We thank the reviewer for his/her appreciation of our work. We amended the experimental section to include the missing procedures.
R. A comment on the comparison of the secondary structure of peptides with and without cells should be added. I don’t see in the CD of spectrum of Leu4-NH2 in buffer or in water. If not present, this should be added.
A. We thank the reviewer for this comment. We added the CD spectrum of Leu4-NH2 in water in the supporting information, acquired with 10% methanol v/v to improve solubility. In the revised version of the manuscript, we have added a comment on the comparison between CD spectra with/without cells, as suggested by this reviewer.
R. One important issue that needs to be clarified is the following: Leu4-GNP are very large, as assessed by DLS measurements. It seems quite strange that such big particles are uptaken by cells. Can the author explain this?
A. As correctly stated by the reviewer, DLS measurements on Leu4-GNPs indicated an average hydrodynamic diameter > 1000 nm, very likely due to the formation of aggregates between particles. On the other hand, as shown in Fig. S13 (reporting TEM images for all NP formulations), the diameter of single Leu4-S-GNPs does not exceed 30-40 nm, i.e. in the same dimension range as that of Api8-S-GNPs. Therefore, it is not surprising that NPs can be taken up by cells (as shown in Fig. 6). In particular, image 12 of Fig. 6 (MDA-MB-231 cells) shows aggregated of Leu4-GNPs interacting with plasma membrane as well as NPs within endocytic vesicles, confirming that, at least in the cell lines considered by us in this study, aggregates of gold NPs are promptly internalized. We added a line in the discussion clarifying this point.
R. The relationship between the results of leakage experiments and the uptake should be better discussed. In addition, it would be desirable to see how K6-FITC performs in membrane leakage and on GNP and also discuss uptake results in comparison with trichogin.
A. We thank the reviewer for the observation. We would have liked to expand the discussion to include the relationship between leakage results and cellular uptake and we indeed tried to perform leakage on FITC-containing peptides. Unfortunately, the fluorescence of FITC covers and interferes with that from CF release, to the point that no conclusions can be drawn from the data collected. Therefore, we perform CD studies on K6-FITC, and added the CD profiles in the supporting information. From the CD profile acquired in the membrane-mimicking environment, we can conclude that the ability to interact with membranes of the peptides is retained by the corresponding, FITC-containing analogs.
R. In Figure 3, please change “lipide” to “lipid”.
A. Thank you for pointing out the typo in the figure. We corrected it in the new version of Fig. 3.
R. Finally, it is clear that a huge amount of work has been performed but presentation of results is in my opinion a little confusing. There are too many information in the tables. As the topic of the paper is the anticancer/cancer cell targeting activity of TRIC analogue the information on PPP in table 1 is not useful. Table 2 could be moved to the Supplementary, as the activity of the peptides is not even reported on the same cell line tested in this work.
A. We thank the reviewer for the observation. We originally reported the activity as plant protection products in Table 1 in response to the topic of the special issue, which was on the activity towards cancer cells of plant-related compounds. In our revised manuscript, we moved the PPP information in the supporting information. We would like to keep Table 2 in the manuscript, since it summarized the antitumor properties of the trichogin analogs studied so far and gives the reader a short but comprehensive picture of the state-of-the-art on the topic, without the need to search the literature on that.

Reviewer 2 Report
The authors provided complex results of the interaction between a natural compound that has high potential in biofunctionalization, which could lead further to targeted treatment of breast cancer.
Aside from several typos (L352, please write the full name of CD, altough it is specified in results, and 408, superscript de 2) the manuscript is well written and it reflects highly important results.
The reviewer agrees with the publication of the manuscript.
Best of luck!
Author Response
We thank the reviewer for his/her positive comments on our work and the useful corrections proposed. We have corrected the typos in the revised version of the manuscript.
Reviewer 3 Report
Nowadays, peptide-nanoparticle (NP) conjugates have been demonstrated to be efficient and powerful tools in the area of drug delivery, targeting system, cancer therapy as well as in bioimaging applications. In this paper, the anticancer activity of natural peptide trichogin analogs from Trichoderma longibrachiatum against a breast cancer cell line and a normal one of the same derivation were studied using different techniques and in vitro tests. The authors described the synthesis and characterization of peptide-decorated gold nanoparticles (GNPs). Circular Dichroism analysis confirms the helical structure of the Api8-NH2 and Leu4-NH2 and the stability of the two analogs in the biological environment. Moreover, cell uptake of the peptide-decorated GNPs was estimated by transmission electron microscopy (TEM) after 24-h incubation. This topic is very interesting, and the obtained results can be further developed in the terms of targeting agents in drug delivery systems. So this paper could be published after a minor revision. The comments are as follows:
-In section “2.2. Synthesis and Characterization of Peptide-decorated gold nanoparticles” I would change the order of Figure S15 and S13, and renumbered them as follows S15ïƒ S13, S13ïƒ S15
- In the section “2.4. Membrane leakage study” I would describe more what method was used to determine CF percentage leakage. This is described in the materials and methods section, but you should add a sentence regarding the fluorescence method in the main section.
- Add conclusion and prospects for future research at the end, after the discussion section.
Author Response
Reviewer: Nowadays, peptide-nanoparticle (NP) conjugates have been demonstrated to be efficient and powerful tools in the area of drug delivery, targeting system, cancer therapy as well as in bioimaging applications. In this paper, the anticancer activity of natural peptide trichogin analogs from Trichoderma longibrachiatum against a breast cancer cell line and a normal one of the same derivation were studied using different techniques and in vitro tests. The authors described the synthesis and characterization of peptide-decorated gold nanoparticles (GNPs). Circular Dichroism analysis confirms the helical structure of the Api8-NH2 and Leu4-NH2 and the stability of the two analogs in the biological environment. Moreover, cell uptake of the peptide-decorated GNPs was estimated by transmission electron microscopy (TEM) after 24-h incubation. This topic is very interesting, and the obtained results can be further developed in the terms of targeting agents in drug delivery systems. So this paper could be published after a minor revision.
Authors: We thank the reviewer for his/her positive comments on our work.
R: The comments are as follows:
-In section “2.2. Synthesis and Characterization of Peptide-decorated gold nanoparticles” I would change the order of Figure S15 and S13, and renumbered them as follows S15>S13, S13>S15
A: Accordingly to reviewer’s suggestion, we have shift the order of Fig. S15 and S13.
R: - In the section “2.4. Membrane leakage study” I would describe more what method was used to determine CF percentage leakage. This is described in the materials and methods section, but you should add a sentence regarding the fluorescence method in the main section.
A. We added a brief description of the method used to determine CF percentage of leakage in section 2.4.
R: - Add conclusion and prospects for future research at the end, after the discussion section.
A. we added a paragraph of Conclusion and Future Perspectives in the revised version of our paper, as suggested by the reviewer.